# Mechanical Characterization of Glued Laminated Beams Containing Selected Wood Species in the Tension Zone

**DOI:** 10.3390/ma15186380

**Published:** 2022-09-14

**Authors:** Adam Derkowski, Marcin Kuliński, Adrian Trociński, Jakub Kawalerczyk, Radosław Mirski

**Affiliations:** Department of Mechanical Wood Technology, Poznań University of Life Sciences, 60-627 Poznań, Poland

**Keywords:** beams, glued laminated timber, modulus of elasticity, pine wood, laboratory tests

## Abstract

The aim of this study was to determine the mechanical properties of laminated beams containing selected wood species in the tension zone using a four-point bending test. Three beam types were manufactured with respect to the timber used in the tension zone, i.e., beams containing oak or beech timber of I and II quality class and pine timber with no defects (as defects had been removed). The manufactured beams were assessed with respect to bending strength and the modulus of elasticity. The obtained results were compared with the performance of BSH (Industrial beams GL made in Germany—Brettschichtholz) industrial beams. We concluded that beams made from pine timber are an appropriate alternative to spruce beams. The static bending strength of the beams made with hardwood faces was 70% higher than that of beams made with pine wood. All types of beams manufactured in the laboratory met the requirements of at least the GL24c class.

## 1. Introduction

The use of wood for construction purposes seems natural both for historical reasons and for the ease with which buildings can be erected, particularly single-family homes. Wood sourced from mature trees (at the felling age) has a sufficient diameter to obtain the required dimensions of construction elements. However, as wood is a natural material, it possesses structural features that can be perceived as defects. Knots, grain slopes, and severe cracks are considered defects of mainly but not exclusively construction timber [1,2,3,4]. Knots are particularly common in construction, and knotless wood is rare. Knots can vary in intensity and size, from very small to the those taking up the entire cross section. Knots considerably impact the mechanical properties of wood, especially with respect to tensile and bending strength [5,6,7,8].

To reduce the impact of naturally occurring wood defects, at least to some extent, the concept of beams made of glued laminated timber, i.e., glulam (GL), has been proposed. This concept assumes homogenization of material by appropriate selection/arrangement of sawn timber in the beam’s cross-section, reducing the impact of negative wood features [9,10,11,12,13,14].

The process of manufacturing construction elements made of glued laminated timber was facilitated by the introduction of system to joining wood pieces with finger joints. This system not only enables the joining of sawn wood of different lengths to produce elements longer than available roundwood pieces but also makes it possible to join pieces of timber from which significant defects have been removed [15,16,17,18]. Exhaustion of load-bearing capacity, especially in the tension zone, resulted in the introduction of various solutions aimed at strengthening the tension zone in industrial practice. The most popular solutions include various types of tapes, rods, or other structures made with glass, carbon, or ballast fibers [19,20,21]. Using even a small percentage of composite materials in the beam structure substantially increases its strength, load-bearing capacity, stiffness, and durability.

Satisfactory results can be also obtained by strengthening wood with steel sections, either in the form of round reinforcing bars or flat bars, [22] or by chemical modification [23]. However, it is much easier to reinforce wood with other wood materials, as opposed to non-wood material. Resorting to non-wood materials requires the use of adhesives, additional operations related to preparing spots for sections, and the use of non-standard saws when shortening such elements. Application of other wood species, e.g., species that exhibit superior mechanical properties, or the use wood-based materials makes it possible to reduce some of these issues. Our previous studies show that satisfactory results can be obtained by using plywood or LVL in the tension zone [24]. Compared with beams made exclusively of pine timber, a substantial increase of approximate 30% in bending strength was obtained for beams manufactured using such a process, in addition to improved elastic properties. However, reinforcement materials are expensive, as well as timber itself, and it is difficult to obtain lamellas that are long enough for such applications, especially in the case of plywood, making it necessary to splice them. Owing to the features of plywood, the process of creating finger joints is much more difficult than making such a joint in wood.

Although coniferous species dominate the northern hemisphere, especially the temperate zone, and they are the basis for manufacturing glued laminated beams, the literature on the subject indicates that positive results can also be obtained using deciduous species. Frese and Blaß [25] proved that glued laminated beams made of beech wood are characterized by a bending strength of more than 44 MPa. Furthermore, Aicher and Stapf [26] demonstrated that beams made of oak exhibited a bending strength exceeding 80 MPa. Other authors also point to the usefulness of hardwood in glulam production [27,28,29,30].

The joining of different species of wood for the manufacture of glued laminated beams with improved mechanical properties is not a new topic. Selected work shows that attempts have already been made to improve the load-bearing characteristics of poplar wood, with precise results. For example, a eucalyptus clone, spruce, and larch were used for the outer layers [31,32]. Other species from the tropics [33] and Europe [34,35] have also been combined.

Lumber with superior mechanical properties is generally placed in the outer zones according to the methodology described in EN-14080 [36]. This standard introduces T grades and provides for the manufacture of beams defined as ‘c’, where the outer layers are made of the higher T grades. With respect to T grades, the standard does not indicate differences in species but only differences in quality. The procedure is only described in general term, without clarity with respect to the possibility of mixing wood species, as evidenced by the publications cited above.

Therefore, the objective of the present study was to determine the potential of using oak and beech timber as an external tension layer in glulam beam structures instead of high-quality timber or pine timber with no defects.

## 2. Methodology

The present research involved pine (*Pinus sylvestris* L.) timber (So), both so-called main timber with a thickness of 40 mm and side timber with a thickness of 25 mm, as well as beech (*Fagus sylvatica* L.) timber (Bk) and oak (*Quercus robur* L.) timber (Db) with a thickness of 28 mm (edged and of the I and II quality classes) (PN-D-96002:1972 [37]). Hardwood timber classified according to this standard is subject to visual assessment. In general, primary defects in wood are taken into account in the evaluation. All defects specified in the standard may be present in each piece of timber, but the surface area free from defects should be: in class I, not less than 80%; in class II, not less than 65%. The sawn timber used in the tests in the present study was 120 mm wide. The hardwood prepared for the research was assessed with respect to modulus of elasticity (MOE) in a four-point bending test with the wood laid flat, as described in [35], and a lever arm equal to 1 m. The distance between supports was also 1 m (Figure 1).

The pine timber was evaluated using the sonic test. According to established timber MOE values, the beam was formed in such a way that sawn timber with a higher modulus of elasticity was placed closer to the outer layer. Beams of 6 or 8 layers were formed. With respect to beams with a face layer made of hardwood, 6 layers of pine timber with a thickness of 40 mm were laid below the face layer, and the bottom layer comprised pine timber with a thickness of 25 mm. The pine timber was 350 cm long, whereas the hardwood was only 315 cm long; therefore, we decided to extend the hardwood by joining the missing part with a finger joint. A FABA PZK 12NS250-002 milling cutter with a pitch of k 10/10 × 3.8 was used to form the joint. Beams made exclusively of pine timber were manufactured as a reference set and were formed with 6 layers. Knots were removed only from the tension zone of the external lamellas, and split elements were joined by finger joints made with the same type of cutter. Immediately before pressing the prepared sets of beams, each piece of timber was planed, resulting in a final thickness 1–1.5 mm less than the initial value.

The beams were cold-pressed with 220–240 g/m^2^ MUF 1247 glue mixed with 2526 hardener in an amount equal to 10% of dry resin mass. Both the glue and the hardener were products of Akzo Nobel (Amsterdam, The Netherlands). The laboratory conditions were taken into consideration when preparing the mixture. Glue was spread using a roller spreader. After loading the press, a pressure of 0.5–0.55 MPa was exerted. Four beams were manufactured every day, with a total of 48 beams were manufactured, i.e., 16 beams of each type. Pressing was performed in an industrial press equipped with hydraulic cylinders dedicated to the production of glued structural elements (FOST, Czersk, Poland).

The beams were pressed for minimum of 4 h, depending on the amount of hardener added to the resin. After two weeks of conditioning, the beams were evaluated for their bending strength and modulus of elasticity in a four-point bending test (Figure 2). The beams were conditioned in a controlled laboratory environment; the temperature was 21 ± 2 °C, and the air humidity ranged from 55 to 65%. A strength testing machine (SAM 500 kN, Poznan, Poland) with a maximum force of 500 kN was used to determine the bending strength and the modulus of elasticity. The device was equipped with a hydraulic drive. The beams were loaded at a speed of 0.20–0.25 mm/s. Moisture content was measured for each beam before it was assessed for its mechanical properties. To this end, resistance a HIT-3 moisture meter was used. The modulus of elasticity was determined by loading the beam eight times with a force (*F*) of about 25 kN, although the strain (*s*) was only recorded for the last five tests. Δ*F* and Δ*s* were determined based on the characteristic curve (*f*(*F*) = *s*), and the *MOE* was determined according to Equation (1):(1)$$MOE=\frac{3\cdot \Delta F\cdot l^{2}\cdot a}{4\cdot b\cdot h^{3}\cdot \Delta s}$$
where Δ*F* is force corresponding to the deflection (N), *l* is the length of the deflection measuring section (mm), *a* is the distance from the applied force to the support (mm), *b* is the beam width (mm), *h* is the beam height (mm), and Δ*s* is deflection (mm).

The *MOE* value for the beam was determined as the average of 5 measurements. An MSL 50.102 PA strain sensor (Larm, Czech Republic) was used in the tests. The strain sensor was then removed from the beam, and force was exerted until the beam collapsed. The *MOR* value was determined according to Equation (2):(2)$$MOR=\frac{3\cdot F_{max}\cdot a}{b\cdot h^{2}}\cdot k_{h}$$
where *F_max_* is the force at which the beam failed (N), and *k_h_* is the coefficient related to the height of the beam.

In addition, the obtained test results were compared with the values exhibited by BSH glued laminated beams purchased in a timber warehouse (12 pieces/beams). These beams were marked as GL24; therefore, according to standard EN 14080 [36], their mean modulus of elasticity would be at least 11.5 GPa, and their bending strength would be at least 24 MPa. The industrial beams had 6 lamellas with a thickness of 39.5 mm, which meant that their size was 120 × 240 mm. However, they were made of spruce (*Picea abies* (L.) H. Karst) timber.

We adopted the following beam arrangements and designations (Figure 3):-G_Db_: 8-layer beam, i.e., 7 layers of pine lumber and 1 of oak lumber;-G_Bk_: 8-layer beam, i.e., 7 layers of pine lumber and 1 of beech lumber;-G_So_: 6-layer beam made from pine lumber only; and-BSH: 6-layer industrial beam made of spruce lumber.

After a period of air conditioning (T = 22 ± 2 °C, RH = 65 ± 5%), the beams were tested for their bending strength and the modulus of elasticity in a four-point bending test. The obtained results were statistically analyzed and compared with the results of previous studies. Statistica software version 13.0 (StatSoft Inc., Tulsa, OK, USA) was used for statistical analysis.

## 3. Results

The quality of lamellas used in this study to manufacture beams in the laboratory is presented in Table 1. The data indicate that the mean value of the modulus of elasticity in the compression zone of the lamellas differs significantly from the mean value of the modulus in the tension zone only in G_Db_ beams. Despite the higher tensile strength of timber along the grain as compared to compression strength, compression is less sensitive to the occurrence of wood defects, which are inevitable in construction wood. Generally, only laboratory-standard samples are free of defects.

The oak and beech timber used in the present study show a similar value of the modulus of elasticity. However, as shown in Figure 4, oak timber is characterized by a very small dispersion of values around the mean value. The results for modulus of elasticity obtained in the present study are consistent with those reported in previous research [38].

Timber for BSH beams with GL24 properties should be characterized by a modulus of elasticity of at least 11 GPa (mean value) if it is from class c (c: combined glulam, EN 14080 [36]), whereas beams from the h group (class h: homogeneous glulam, EN 14080 [36]) should have an MOE of at least 11.5 GPa. By removing the defects from the purchased pine timber, we increased the elastic modulus from about 11 GPa to about 13 GPa. It can be assumed that a similar situation will occur with the spruce lumber used for BSH. Consequently, both types of beams, i.e., industrial (BSH) and the laboratory (GSo), should show similar elastic properties and meet the standard of at least GL26.

According to the data shown in Table 2, the mean strength of the tested beams is quite high. The lowest value was noted for industrial beams (BSH, 25.5 MPa). The strength of the beams made of the pine timber is almost 6.5 MPa higher, whereas that of beams with a hardwood face is nearly 40% lower. Although the static bending strength of the beams made with oak timber is 4.2 MPa lower than that of beams made with beech timber, there is no statistical difference in strength between these two types (Table 2, column 2). The beams with faces made of hardwood timber show left-skewed distribution, which means that deviating values are lower than the mean value, as opposed to beams with faces made of softwood timber. These, in turn, show a right-skewed distribution. Only BSH beams exhibit positive kurtosis, which means that they are the set of beams with the highest variation coefficient.

It is not the mean value but the characteristic value or the value for the fifth percentile that determines the assignment of a given batch of beams to a particular quality class with respect to static bending strength. In this case, beams with hardwood faces can be classified as GL32h (h: homogeneous glulam, EN 14080 [36]). Homogeneous glulam is a high-quality class, given there was only one hardwood lamella in each beam, and the share of its thickness in the cross section/height of the beam was less than 9%.

The strength of the pine beams (G_So_) is at the level of 24.6 MPa, so they should be classified as GL24c. In this case, the class is lower than expected. Our previous research showed that beams made of pine timber exhibit a strength of 24–32 MPa, depending on the quality of the tension zone [39]. Industrial beams do not meet the requirements of the class in which they are classified. Their mean strength is satisfactory, although the value for the 5th percentile falls significantly below expectations (24 MPa). The tested batch of beams cannot be assigned to any class. As is presented in Figure 5 as many as 5 beams of 12 have a strength lower than expected. One beam had a strength of almost 42 MPa, representing a high mean value for this batch of beams. Only 5 of 12 beams failed as a result of damage to external lamellas at the location of finger joints (Figure 6, top photo). In all other cases, the cause of damage was the quality of external lamellas in the tension zone (Figure 6, bottom photo). It the opposite of what was observed for the laboratory beams made of pine timber (G_So_), as in the case of these beams, the main reason for failure was the destruction of finger joints (Figure 7).

In contrast, beams with hardwood faces are not only characterized by smaller distribution but also by a 35% higher strength. The inclusion of high-quality beech and oak timber ensures strength in this case. As shown Figure 8 and Figure 9, the beams were damaged because the core pine lamellas lost their bearing capacity.

Face lamellas made of oak or beech were not damaged. This was true for all G_Db_ and G_Bk_ beams, indicating that the full potential of the hardwood face was not used. However, no wood defects were removed here from pine timber; therefore, it would be necessary to use high-quality or defect-free pine timber to obtain improved bearing capacity.

The manufactured beams, despite a high bending strength—much higher than that of industrial beams—have a modulus of elasticity similar to that of industrial BSH beams. As shown in Figure 10, no statistical differences were observed in the elasticity of BSH, G_So_, and G_Db_ beams. The mean value of the modulus obtained for these ranges from 11.0 GPa to 11.3 GPa. These are only the beams with a beech timber face to exhibit a modulus of elasticity higher than about 1 GPa. The results obtained for G_Db_ and G_So_ beams are lower than expected, as data displayed in Figure 4 indicate that their MOE should be 11.9 GPa and 11.4 GPa, respectively. The results obtained for the two other types of beams, i.e., BSH and G_Bk_, are as expected. The EN 14080 [36] standard requires verification of the values for the fifth percentile. In this case, BSH beams fail as in the bending test, as they do not exhibit sufficient stiffness. The required value for the fifth percentile is 9.1 GPa, whereas these beams demonstrated a value of only 8.6 GPa. The modulus of elasticity both for timber and glued laminated elements depends heavily on moisture content. Because wood is highly hygroscopic, it is recommended to test its mechanical properties at 12% moisture. Conditioning large elements to achieve the same moisture content in all the cross sections is difficult; therefore, correction formulae were applied, e.g., the Bauschinger formula [39]. The moisture content of the tested G_Db_ and G_Bk_ beams is around 12%, differing only by 0.2% (Figure 11). The moisture content of the BSH and G_So_ beams is slightly lower i.e., around 11%. The standard does not provide any explicit guidelines with respect to how to recalculate results with wood moisture content lower than 12%. Only coefficients (α_MC_: coefficient of variation of the modulus of elasticity of wood after its moisture content changed by 1%) of 0.04, 0.02, and 0.01 are included in the standard. Because the differences in moisture content of the beams and the normative moisture are low, recalculation of the values obtained in the test “normative” values was abandoned. Recalculation would surely result in lower values than those presented in Figure 10; however, it would not affect the GL classification of the beams.

## 4. Conclusions

The following conclusions can be drawn based on the obtained results:-The tension zone can be reinforced with hardwood timber or with pine timber free of defects;-Laboratory beams made with a hardwood face exhibited a strength 70% higher than that of laboratory beams made of pine timber only;-Laboratory beams made of pine timber show higher technical parameters (MOR and MOE) than industrial beams made of spruce timber;-The tested beam types confirmed the assumptions for GL24 with respect to their mean modulus of elasticity;-Generally, the obtained values of modulus of elasticity are slightly lower than those estimated based on the modulus of elasticity of the timber itself; -When designing beam stiffness based on the modulus of elasticity of timber itself, we recommend designing beams to have a modulus of elasticity approximately 0.5 GPa higher than the ultimately expected value.

## Figures and Tables

**Figure 1 materials-15-06380-f001:**
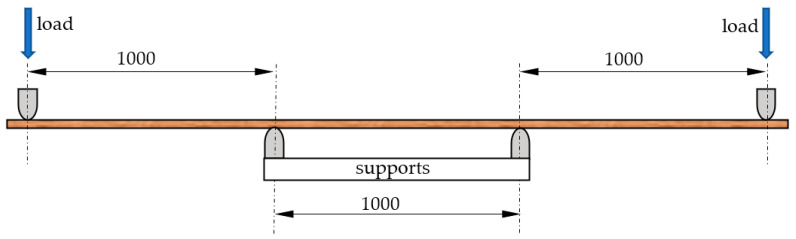
Schematic of the testing stand for the MOE of sawn timber.

**Figure 2 materials-15-06380-f002:**
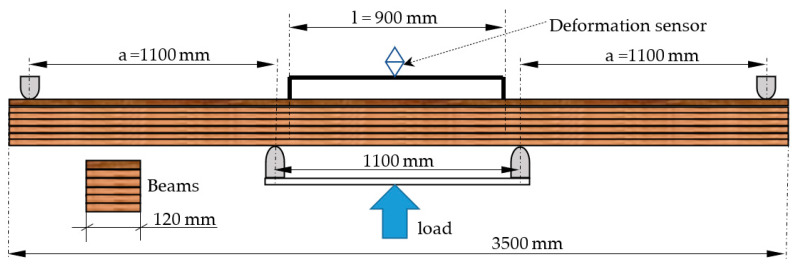
Schematic of the tested beam.

**Figure 3 materials-15-06380-f003:**
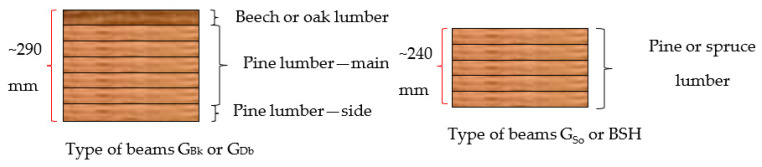
Diagrams of the arrangement of sawn timber in beams.

**Figure 4 materials-15-06380-f004:**
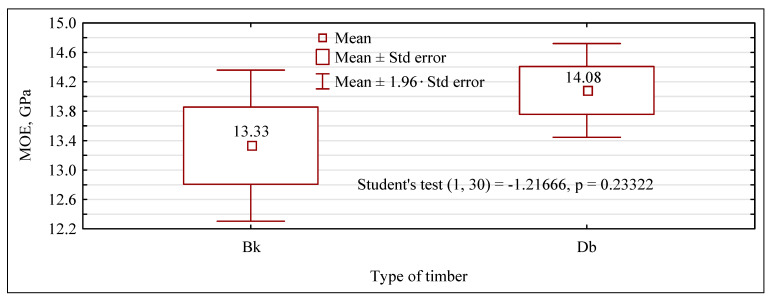
Oak (Db) and beech (Bk) timber.

**Figure 5 materials-15-06380-f005:**
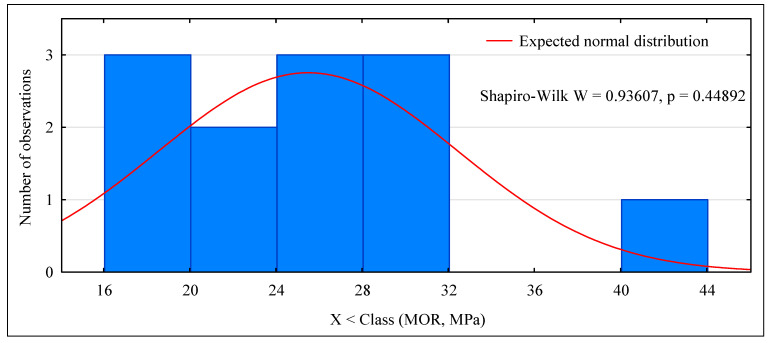
Histogram of static bending strength for BSH beams.

**Figure 6 materials-15-06380-f006:**
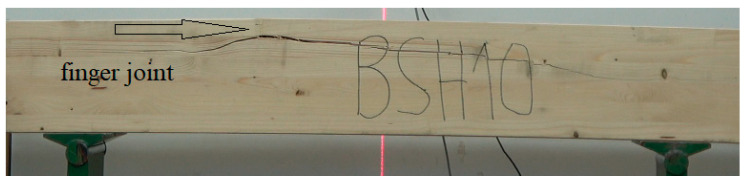
Propagation of damage in BSH beams.

**Figure 7 materials-15-06380-f007:**
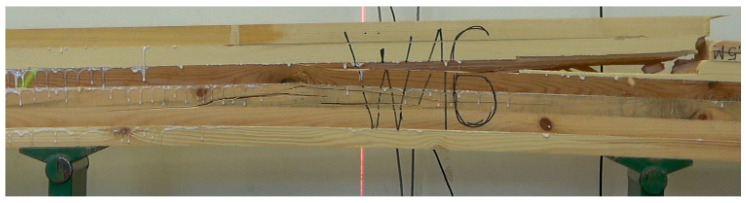
Propagation of damage in G_So_ beams.

**Figure 8 materials-15-06380-f008:**
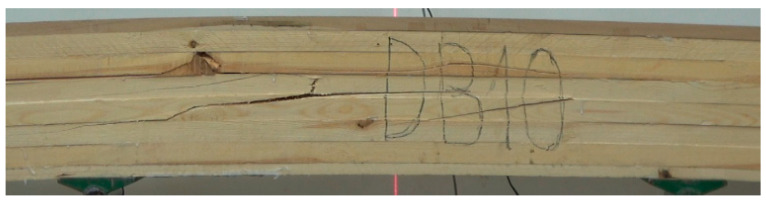
Propagation of damage in G_Db_ beams.

**Figure 9 materials-15-06380-f009:**
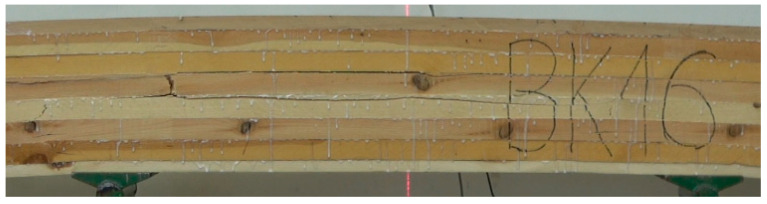
Propagation of damage in G_Bk_ beams.

**Figure 10 materials-15-06380-f010:**
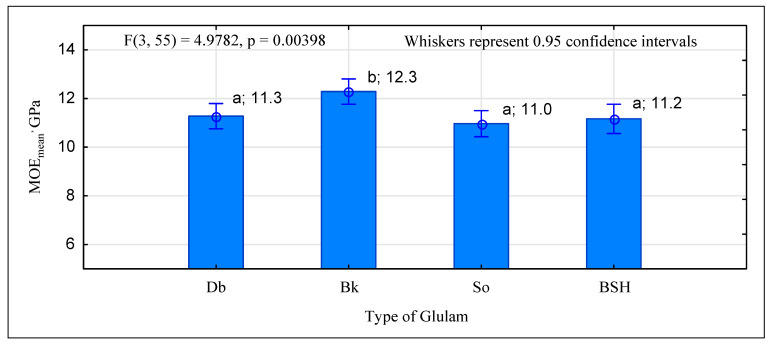
Statistical analysis of the modulus of elasticity for the tested beam types. Small letters a, b indicate homogenous groups based on Tukey’s test.

**Figure 11 materials-15-06380-f011:**
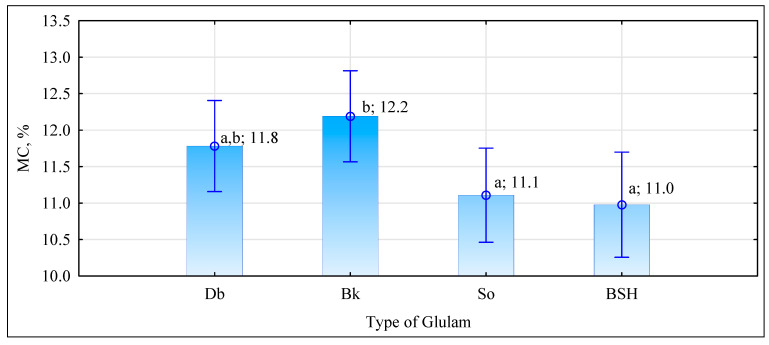
Mean moisture content of the tested beam types. Small letters a, b indicate homogenous groups based on Tukey’s test.

**Table 1 materials-15-06380-t001:** Values of the modulus of elasticity (MOE) for the lamellas used in the study.

Type	Feature	Lam. 1 *	Lam. 2	Lam. 3	Lam. 4	Lam. 5	Lam. 6	Lam. 7	Lam. 8
G_So_	MOE (SD **), GPa	13.0 (0.149)	11.25 (0.868)	9.21 (0.677)	9.31 (0.686)	11.14 (0.889)	12.45 (0.543)	-	-
G_Db_	MOE (SD), GPa	14.08 (1.261)	12.93 (0.643)	11.02 (0.442)	9.22 (0.608)	9.24 (0.593)	11.01 (0.448)	12.90 (0.648)	10.59 (1.938)
G_Bk_	MOE (SD), GPa	13.33 (2.030)	12.86 (0.751)	11.02 (0.487)	9.28 (0.609)	9.31 (0.580)	11.00 (0.484)	12.84 (0.753)	13.14 (2.930)

* Lam. 1 indicates the outer lamella of the tension zone (the number next to lam. indicates the consecutive number of the lamellas); ** SD, standard deviation.

**Table 2 materials-15-06380-t002:** Static bending strength (MOR) of the beams used in the tests.

Type of Beam	Mean Value, MPa	ν, %	Skewness	Kurtosis	5% Percentile MPa
G_Db_	47.85 ^a^	7.8	−0.46	−0.88	40.67
G_Bk_	52.07 ^a^	7.8	−0.26	−0.22	43.98
G_So_	31.92 ^b^	16.4	0.62	−0.20	24.61
BSH	25.50 ^c^	27.3	0.94	1.65	16.05

Small letters indicate homogenous groups according to the Tukey TSH test.

## Data Availability

The data presented in this study are available on request from the corresponding author.

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
