# Peer review of "Mechanical Characterization of Glued Laminated Beams Containing Selected Wood Species in the Tension Zone"

_materials, 2022, doi:10.3390/ma15186380_

Round 1

Reviewer 1 Report

Dear auhtors

The suggestion and comment can be seen in attachement. 

Author Response

Thank you very much for your careful review. It will not only improve the quality of this publication but also enhance the quality of our research skills.

Commented [M1]: What species? Suggest changing it as selected wood species. And indicate the loading conditio

Answer: Corrected in the text

Commented [M2]: The abstract should be rewritten to clearly indicate the aim. content and results of this study.

Answer: Corrected in the text

Commented [M3]: Full spell it when first appear

Answer: Corrected in the text

Commented [M4]: It seems a common sense that hardwood can enhance the tension strength of beam made of lightweight wood species.

Answer: Indeed, this seems to be the case. However, according to our as yet unpublished research, the problem is more complex.

Commented [M5]: The introduction section should be separated into several paragraphs to make it logical and structural

Answer: Corrected in the text

Commented [M6]: chemical modification also a commonly used method to enhance the wood, such as doi:10.3390/f11050551

Answer: Corrected in the text

Commented [M7]: This writing style is unreadable. This part must be including configurations of samples, testing methods, and experimental design. Separate it into the three parts, and use diagrams to indicate the dimensions and layout of sample, as well as photos to indicate testing method.

Answer: Corrected in the text

Commented [M8]: I think you forgot deleting them.

Answer: Corrected in the text

Commented [M9]: Are there any discussions in this paper? Also the typical load-deflection curves should be added and discussed. See DOI: 10.15376/biores.16.4.7101-7111

Answer: It is there, but poor indeed, as it only refers to our research. Thank you for pointing out the publication. I will include it in our next paper on beams or CLTs.

Commented [M10]: Full spell it. Also Gso and Gbk.

Answer: Corrected in the text

Commented [M11]: It is unreadable in this form. Just as i recommended you must indicate the beams you studied including dimensions and layout using diagrams

Answer: Corrected in the text

Commented [M12]: [Student’s t-test] in figure 1--- What does it mean?

Answer: Student’s t-test, in statistics, a method of testing hypotheses.

Commented [M13]: Lots of shorts were not indicated, full spell them. Thoroughly check through the paper.

Answer: Corrected in the text

Commented [M14]: The reference values should be indicated.

Answer: To my mind, specifying the GL24c value indicates which bending strength is referred to.

Commented [M15]: Add reference value.

Answer: Corrected in the text

Commented [M16]: How many samples did you test? The normal distribution of data should be conducted with enough samples. in Methodology section, it seems that the replication is not satisfied this analysis

Answer: We only tested 12 beams. Indeed, the sample is not numerous, and perhaps it cannot be applied to the production as a whole. However, from the user's point of view, the low values obtained are a significant indication that there is something wrong with the quality of the production. We only used Statistica software.

Commented [M17]: The number should be No. 4.

 Answer: Corrected in the text

Commented [M18]: This paragraph is background and aim of this study. Replication of this is not necessary

Answer: Corrected in the text

Reviewer 2 Report

some minor corrections are necessary

Author Response

Thank you very much for your favourable review. We have made the suggested corrections to the text.

Reviewer 3 Report

My recomendation is: minor revision. The paper is interesting and presents possibility of using hardwoods for beams.

Author Response

Thank you very much for your careful review. It will not only improve the quality of this publication but also enhance the quality of our research skills.

Commented [M1]: How many layers?

Answer: Corrected in the text. The outer zone was only one lamella/layer.

Commented [M2]: Does that refer to the beam or?

Answer: No, it applies generally to the tree. Corrected in the text.

Commented [M3]: Latin name.

Answer: Corrected in the text.

Commented [M4]: Latin name.

Answer: Corrected in the text.

Commented [M5]: Latin name.

Answer: Corrected in the text.

Commented [M6]: Why bending test for hardwood and sonic for pine? How MoE was determined, at bending test; what was the starting point of determination and what was the end point?

Answer: We bought only sonically sorted pine lumber. At that time, we did not have yet a sonic test stand, so we determined the MOE of sawn timber in a bend test. We supplemented the information in the methodology.

Commented [M7]: Why such structure/configuration?

Answer: Corrected in the text.

Commented [M8]: At least?

Answer: Corrected in the text.

Commented [M9]: At least?

Answer: Corrected in the text.

Commented [M10]: statistic or statistical analysis

Answer: Thanks for the hint - corrected in the text.

Commented [M11]: This section probably needs to be deleted.

Answer:  Thanks for the hint - corrected in the text.

Commented [M12]: is

Answer: Corrected in the text.

Commented [M13]: GDb bemas are... I recomend that in the Methodology sectio you show the configuration of beams with their marking in separate table.

Answer:  Corrected in the text.

Commented [M14]: Is this correct? What is the meaning of asterisk?

Answer: Corrected in the text.

Commented [M15]: How, with the narrow side oriented how, upwards, on the supports?

Answer: Corrected in the text.

Commented [M16]: Which previous research?

Answer: Corrected in the text.

Commented [M17]: h group - meaning?

Answer: Corrected in the text.

Commented [M18]: What does GL26c class means; what is the difference between GL26 and GL26c?

Answer: The EN 14080 standard divides glulam into two groups: c - combined glulam and h - h - homogeneous glulam. The groups differ in some mechanical properties. The differences relate essentially to the modulus of elasticity and tensile strength. For both thicknesses, the bending strength is constant and it is given the number after GL.

Commented [M19]: Meaning? Limit values?

Answer:  GL32h is a glulam with a characteristic static bending strength of 32 MPa (N/mm2) made of lamellas of the same T class.

Reviewer 4 Report

The paper deals with the mechanical characterization of homogeneous and mixed (softwood / hardwood) glue laminated timber. The work could be useful to add data to a topic of current interest. To this aim however, it needs to be deeply improved: the test methods and the calculations should be clearly stated to make future comparisons possible.

My general comments are as follows:

English: the language should be improved for many sentences are not very clear to the reader.

Title: it does not reflect the content of the work, I would better speak of mechanical characterization than investigating “the possibility” of using hardwoods.

Introduction: The scientific literature of previous works describing the use of a mix of species in the production of glulam is completely missing (some examples here: Castro et al 2013 doi: 10.1007/s00107-003-0393-6; Rescalvo et al 2020 doi: 10.3390/ma13143134; Sciomenta et al 2022 doi: 10.1016/j.engstruct.2022.114450). At the same time, the part dedicated to the reinforcement of glulam beams with fabrics or rods is too long and the number of cited works too high, for it is not the focus of the paper.

Moreover, the use of stronger and stiffer timber in the outer layers to rise the mechanical performance of glulam beams is well established, so to be included in the European standard EN 14080 (cited by the authors) as combined glulam. In the standard it is not a mixture of species, but a mixture of strength classes, but it should be mentioned in the introduction.

Methods: The methods used should be very clear and exhaustive to allow for future comparisons. As an example: the test geometry used; how the modulus of elasticity was measured; the calculations of the characteristic values…

Captions, symbols and acronyms: Captions of tables and figures need to be much more informative! They should be read independently from the text. Symbols and acronyms should be explained clearly.

Results and conclusions: statements are often inaccurate and approximate, please, take care to be more rigorous.

Specific comments:

Lines 30-32 and 35-37. not clear, please rephrase.

L 77-79. main timber is the one used in the inner layers? and edge in the outer layers?

L 79. what does quality class I and II mean here? Was the timber strength graded?

L 81-82. The test geometry was not the one of EN 408? Why not? Why do not measure the dynamic modulus as for pine?

L 83-84. not clear, please rephrase.

L 85-87. a table or a scheme of the layout tested would improve the readability of the work.

L 91. It seems that the specimens had different size (height). Did the author apply any size factor to make the strength results comparable?

L 105. Four point bending test: please describe the geometry of the test and how the modulus of elasticity was measured and calculated

L 113. How many BSH (describe the meaning of the acronym, by the way) were tested?

L 116. Their “characteristic” strength should be at least 24 MPa.

L 119. “after the period of exerting bending load” what does it mean?

L 124 – 131. Delete.

L 134. GDb should be explained at least the first time you use it.

Table 1. Caption is not correct (not only mean values are displayed) and not exhaustive (acronyms and symbols need to be explained). Moreover, what was the “Lam.” (layer?) stressed in tension and what in compression? Why some SD are missing (Lam 1)?

L 145-147. From what the authors deduce it? By the way, the EN 408 does not require to test the pieces edgewise.

Figure 1. Again the caption. What Bk and Db means?

L 155. The “mean” modulus is 11 GPa.

L 155. H group refers to homogeneous glulam of EN 14080, I presume. Please, be clear!

L 157-159. What the authors refer to?? Spruce of BSH or pine?? Again, be clear!

Table 2. caption: what does “statistical properties” mean? Mean values? CV? And what is TSH on the not? Tukey test??

L 177 and further on. How the 5-percentile was calculated? There are several methods to calculate it!

L 248-261. This is part of the introduction!

L 264-265. It is not an outcome of the present work

L 270-271. What laboratory beams? There are several type of specimens.

L 272-273. What assumptions?

Author Response

Thank you very much for your careful review. It will not only improve the quality of this publication but also enhance the quality of our research skills.

My general comments are as follows:

English: the language should be improved for many sentences are not very clear to the reader.

Answer: We've improved some things

Title: it does not reflect the content of the work, I would better speak of mechanical characterization than investigating “the possibility” of using hardwoods.

Answer: Corrected in the text.

Introduction: The scientific literature of previous works describing the use of a mix of species in the production of glulam is completely missing (some examples here: Castro et al 2013 doi: 10.1007/s00107-003-0393-6; Rescalvo et al 2020 doi: 10.3390/ma13143134; Sciomenta et al 2022 doi: 10.1016/j.engstruct.2022.114450). At the same time, the part dedicated to the reinforcement of glulam beams with fabrics or rods is too long and the number of cited works too high, for it is not the focus of the paper.

Answer: Thank you for your valuable remarks. Corrected in the text.

Moreover, the use of stronger and stiffer timber in the outer layers to rise the mechanical performance of glulam beams is well established, so to be included in the European standard EN 14080 (cited by the authors) as combined glulam. In the standard it is not a mixture of species, but a mixture of strength classes, but it should be mentioned in the introduction.

Answer: Thank you for your valuable remarks. Corrected in the text.

Methods: The methods used should be very clear and exhaustive to allow for future comparisons. As an example: the test geometry used; how the modulus of elasticity was measured; the calculations of the characteristic values…

Answer: Thank you for your valuable remarks. Corrected in the text.

Captions, symbols and acronyms: Captions of tables and figures need to be much more informative! They should be read independently from the text. Symbols and acronyms should be explained clearly.

Answer: Thank you for your valuable remarks. Corrected in the text.

Results and conclusions: statements are often inaccurate and approximate, please, take care to be more rigorous.

Answer: Thank you for your valuable remarks. Corrected in the text.

Specific comments:

Lines 30-32 and 35-37. not clear, please rephrase.

Answer: Corrected in the text.

L 77-79. main timber is the one used in the inner layers? and edge in the outer layers?

Answer: No, side lumber only as thickness correction in type beams GBd, GBk. Corrected in the text.

L 79. what does quality class I and II mean here? Was the timber strength graded?

Corrected in the text.

L 81-82. The test geometry was not the one of EN 408? Why not? Why do not measure the dynamic modulus as for pine?

Answer: We decided that we would study the lumber as it works in beams. The project concerned pine lumber. We tested the lumber flat at our institution and at the Warsaw university according to EN 408. (Burawska-Kupniewska, I., Krzosek, S., Mankowski, P., Grzeskiewicz, M., and Mazurek, A. (2019). "The influence of pine logs (Pinus sylvestris L.) quality class on the mechanical properties of timber," BioRes. 14(4), 9287-9297). We found no significant differences in MOE for the two methods. We bought pine lumber already with a specific MOE. In contrast, neither we nor the sawmill from which we purchased the hardwood lumber had measuring equipment for MOE by the sonic method.

L 83-84. not clear, please rephrase.

Answer: Corrected in the text.

L 85-87. a table or a scheme of the layout tested would improve the readability of the work.

Answer: Corrected in the text.

L 91. It seems that the specimens had different size (height). Did the author apply any size factor to make the strength results comparable?

Answer: Yes, the beams had different heights. We considered it difficult to see this variability with the high variability of other factors.

L 105. Four point bending test: please describe the geometry of the test and how the modulus of elasticity was measured and calculated

Answer: Corrected in the text.

L 113. How many BSH (describe the meaning of the acronym, by the way) were tested?

Answer: Corrected in the text.

L 116. Their “characteristic” strength should be at least 24 MPa.

Answer: Corrected in the text

L 119. “after the period of exerting bending load” what does it mean?

Answer: Corrected in the text (After a period of air conditioning)

L 124 – 131. Delete.

Answer: Corrected in the text

L 134. GDb should be explained at least the first time you use it.

Answer: Corrected in the text

Table 1. Caption is not correct (not only mean values are displayed) and not exhaustive (acronyms and symbols need to be explained). Moreover, what was the “Lam.” (layer?) stressed in tension and what in compression? Why some SD are missing (Lam 1)?

Answer: Corrected in the text

L 145-147. From what the authors deduce it? By the way, the EN 408 does not require to test the pieces edgewise.

Perhaps the latest standard does not require it. However, it defines, which is sometimes challenging to meet, the minimum support spacing as 18h ± 3h, which means that the larger the h, the longer the element we can examine. Perhaps the differences we see are just an error in the method, but these are our assumptions.

 „h depth of cross section in a bending test, or the larger dimension of the cross section, in millimetres” According to  EN 408:1995 Timber structures — Structural timber and glued laminated timber — Determination of some physical and mechanical properties published by the European Committee for Standardization (CEN)

Figure 1. Again the caption. What Bk and Db means?

Answer: Corrected in the text

L 155. The “mean” modulus is 11 GPa.

Answer: Corrected in the text

L 155. H group refers to homogeneous glulam of EN 14080, I presume. Please, be clear!

Answer: Corrected in the text

L 157-159. What the authors refer to?? Spruce of BSH or pine?? Again, be clear!

Answer: Corrected in the text

Table 2. caption: what does “statistical properties” mean? Mean values? CV? And what is TSH on the not? Tukey test??

Answer: Corrected in the text

L 177 and further on. How the 5-percentile was calculated? There are several methods to calculate it!

Honestly, I didn't think about it - I used the Statistica program.

L 248-261. This is part of the introduction!

Answer: Corrected in the text

L 264-265. It is not an outcome of the present work

Answer: Corrected in the text

L 270-271. What laboratory beams? There are several type of specimens.

Answer: Corrected in the text

L 272-273. What assumptions?

Answer: Corrected in the text

Round 2

Reviewer 1 Report

Dear authors

The paper has been much improved, especially the experimetal design. Please also check the terms and shorts in the paper. 

Author Response

Dear Reviewer

Thank you very much for your help in preparing our article. We've made a few more minor changes that should make it even better.

Reviewer 4 Report

Generally speaking, the revision was not very thorough. Obviously, the comments and suggestions of a reviewer can be argued, but it should not go on writing in the answers that something was corrected in the text when it was not.

Anyhow, my main concern remains the superficiality with which the authors approach the various aspects of the mechanical characterization of structural products: the size effect; the effect of the span during the test; the calculation of the characteristic values; not specifying how the MOE was measured (the global modulus contain the shear effect, which, with this very short span is emphasized; the dynamic modulus is usually higher than the static one, but the authors did not considered this aspect neither ….).

If the comparison is between specimens with the same geometric characteristics and tested in the same way, deviations from the standards can be accepted (even if the data provided lose interest for future comparisons with other laboratories), but care must be taken to compare data contained in the technical regulations which are based on different standardized methodologies.

Attached is a word file with the previous comments, where I replied the authors' answers. 

Author Response

Dear reviewer

Thank you very much for your help in preparing our article. We've made a few more minor changes that should make it even better.

Detailed comments are available in the attached file.
